# Ceftazidime–Avibactam in Critically Ill Patients: A Multicenter Observational Study

**DOI:** 10.3390/antibiotics14080797

**Published:** 2025-08-05

**Authors:** Olivieri Silvia, Sara Mazzanti, Gabriele Gelo Signorino, Francesco Pallotta, Andrea Ficola, Benedetta Canovari, Vanessa Di Muzio, Michele Di Prinzio, Elisabetta Cerutti, Abele Donati, Andrea Giacometti, Francesco Barchiesi, Lucia Brescini

**Affiliations:** 1Infectious Diseases, Via Murri, 15, 63023 Fermo, Italy; silvia.olivieri2@sanita.marche.it; 2Infectious Diseases Clinic, Via Conca 71, 60126 Torrette, Italy; sara.mazzanti@ospedaliriuniti.marche.it (S.M.); gabrielegelo@outlook.it (G.G.S.); andrea.ficola3@gmail.com (A.F.); a.giacometti@staff.univpm.it (A.G.); 3Department of Biomedical Sciences and Public Health, Medical School, Polytechnic University of Marche, Via Tronto 10/A, 60020 Torrette, Italy; f.pallotta@univpm.it (F.P.); f.barchiesi@staff.univpm.it (F.B.); 4Infectious Diseases Unit, Azienda Ospedaliera Ospedali Riuniti Marche Nord, 61121 Pesaro, Italy; benedettacanovari@ospedalimarchenord.it; 5Hospital Pharmacy, AUO delle Marche, Via Conca 71, 60126 Torrette, Italy; vanessa.dimuzio@ospedaliriuniti.marche.it (V.D.M.); michelediprinzio@ospedaliriuniti.marche.it (M.D.P.); 6Department of Anesthesia, Transplant and Surgical Intensive Care, AUO delle Marche, Via Conca 71, 60126 Torrette, Italy; 7Anesthaesia and Intensive Care Clinic, AUO delle Marche, Via Conca 71, 60126 Torrette, Italy

**Keywords:** critically ill patients, ceftazidime/avibactam, liver transplantation

## Abstract

Ceftazidime–avibactam (CAZ-AVI) is a second-generation intravenous β-lactam/β-lactamase inhibitor combination. In recent years, substantial evidence has emerged regarding the efficacy and safety of CAZ-AVI. However, data on its use in critically ill patients remain limited. Background/Objectives: This multicenter, retrospective, observational cohort study was conducted across four Intensive Care Units (ICUs) in three hospitals in the Marche region of Italy. The primary objective was to evaluate the 30-day clinical outcomes and identify risk factors associated with 30-day clinical failure—defined as death, microbiological recurrence, or persistence within 30 days after discontinuation of therapy—in critically ill patients treated with CAZ-AVI. Methods: The study included all adult critically ill patients admitted to the participating ICUs between January 2020 and September 2023 who received CAZ-AVI for at least 72 h for the treatment of a confirmed or suspected Gram-negative bacterial (GNB) infection. Results: Among the 161 patients included in the study, CAZ-AVI treatment resulted in a positive clinical outcome (i.e., clinical improvement and 30-day survival) in 58% of cases (n = 93/161), while the overall mortality rate was 24% (n = 38/161). Relapse or persistent infection occurred in a substantial proportion of patients (25%, n = 41/161). Notably, acquired resistance to CAZ-AVI was observed in 26% of these cases, likely due to suboptimal use of the drug in relation to its pharmacokinetic/pharmacodynamic (PK/PD) properties in critically ill patients. Furthermore, treatment failure was more frequent among immunosuppressed individuals, particularly liver transplant recipients. Conclusions: This study demonstrates that the mortality rate among ICU patients treated with this novel antimicrobial combination is consistent with findings from other studies involving heterogeneous populations. However, the rapid emergence of resistance underscores the need for vigilant surveillance and the implementation of robust antimicrobial stewardship strategies.

## 1. Introduction

Hospital-acquired infections in critically ill patients are predominantly caused by Gram-negative bacteria (GNB) [1], which are associated with high morbidity, mortality, and healthcare costs [2]. The increasing prevalence of multidrug-resistant (MDR) pathogens—such as carbapenem-resistant *Enterobacterales* (CRE), *Pseudomonas aeruginosa*, and *Acinetobacter baumannii*—has drastically limited therapeutic options. Italy, in particular, bears one of the highest burdens of antimicrobial resistance in Europe [3,4]. ICU-acquired infections, especially those caused by carbapenem- and cephalosporin-resistant strains, are independently associated with increased mortality, as confirmed by large-scale epidemiological studies [5].

The primary mechanism of resistance in these pathogens is the production of β-lactamases, including KPC, OXA, and metallo-β-lactamases (MBLs), which render many β-lactam antibiotics ineffective [6,7]. This resistance is further exacerbated by factors common in ICU settings, such as prolonged hospital stays, invasive procedures, and the overuse of broad-spectrum antibiotics—particularly during the COVID-19 pandemic [8,9,10,11,12]. Traditional agents like polymyxins and aminoglycosides have limited efficacy and safety, prompting the urgent need for novel therapeutic options.

Ceftazidime–avibactam (CAZ-AVI) is a novel β-lactam/β-lactamase inhibitor combination with activity against class A, C, and some class D β-lactamases, including KPC-producing strains [13]. Approved for various complicated infections, CAZ-AVI has rapidly become a key treatment for CRE infections. However, data on its use in critically ill patients remain limited. This study aims to evaluate the clinical outcomes of CAZ-AVI in ICU patients and to identify risk factors for treatment failure, with particular attention to resistance development. Additionally, a sub-analysis was performed in liver transplant recipients—a clinically vulnerable subgroup at high risk for MDR infections due to multiple factors—for whom data on CAZ-AVI use remain limited and inconclusive.

## 2. Results

During the study period, a total of 161 critically ill patients received CAZ-AVI therapy. The majority were male (68%), with a median age of 64 years (IQR 56–73). At ICU admission, 85% of patients presented with at least one comorbidity: cardiovascular diseases (54%), diabetes mellitus (22%), neurological disorders (18%), gastrointestinal disorders (17%), chronic kidney disease (CKD) (15%), chronic obstructive pulmonary disease (COPD) (15%), oncological disorders (8%), hematological diseases (7%), and solid-organ transplantation (SOT) (9%). The Charlson Comorbidity Index (CCI) was greater than 3 in 64% of patients.

Thirty-five percent of patients (57/161) were admitted to the ICU for postoperative care, 33% (53/161) for acute respiratory failure, 14% (23/161) for polytrauma, 7% (12/161) for sepsis or septic shock, and 12% (20/161) for other reasons, including necrotic hemorrhagic pancreatitis, hepatic decompensation, stroke, status epilepticus, cardiocirculatory arrest, and encephalitis. Additionally, 34% (54/161) were affected by severe SARS-CoV-2-related pneumonia.

A total of 32% of patients (51/161) were transferred from a semi-intensive or intensive care unit, 27% (43/161) from a medical ward, and 16% (26/161) from a surgical ward.

The median APACHE II score at ICU admission was 11 (IQR 8–14.5), while the median SOFA score was 5 (IQR 4–7).

Ventilator-associated pneumonia (VAP) was the most common infection requiring CAZ-AVI treatment, accounting for 54% of cases. Bloodstream infections (BSIs) and complicated urinary tract infections (cUTIs) were the second most frequent indications, each representing 15% of cases. CAZ-AVI was administered for complicated intra-abdominal infections (cIAIs) in 4% of patients, and for suspected MDR GNB infections with limited treatment options in 12%. In 4% of cases, CAZ-AVI was prescribed for catheter-related bloodstream infections (CLABSIs), while in 3%, it was used for other indications, such as surgical site infections.

A total of 53% of patients (86/161) had been hospitalized for more than one month prior to the onset of infection, and 24% (39/161) had a documented rectal carriage of carbapenem-resistant *Enterobacterales* (CRE) within six months before ICU admission.

A total of 75% of patients (121/161) developed acute complications during their ICU stay, including septic shock in 35% (57/161), acute kidney injury in 20% (32/161), acute respiratory distress syndrome (ARDS) in 27% (43/161), bleeding in 16% (26/161), and pneumonia in 65% (104/161).

Invasive devices were frequently used: 81% (130/161) had a central venous catheter, 77% (124/161) had a nasogastric tube, 86% (139/161) had a bladder catheter, 30% (48/161) had one or more surgical drains, and 24% (39/161) had a tracheostomy within 72 h prior to infection onset.

Most patients required life-supporting therapies: 83% (133/161) received mechanical ventilation, 35% (57/161) required vasopressors, 17% (27/161) underwent renal replacement therapy, and 8% (13/161) were treated with extracorporeal membrane oxygenation (ECMO).

The microbiological characteristics of infections treated with CAZ-AVI are summarized in Table 1. *K. pneumoniae* was the most frequently isolated pathogen (47% of isolates), followed by *P. aeruginosa* (32%). Eight percent of infections were polymicrobial.

Among non-*Pseudomonas* isolates, 44% were classified as multidrug-resistant (MDR), including 37% producing KPC carbapenemases, 1% producing OXA-48 carbapenemases, and 8% producing extended-spectrum β-lactamases (ESBLs). Among *P. aeruginosa* isolates, 24% were wild-type strains, 7% were MDR, and 1% were classified as difficult-to-treat resistant (DTR).

CAZ-AVI was administered as monotherapy in 75% of cases, while 24% of patients received CAZ-AVI in combination with at least one other in vitro active antibiotic. The most frequently co-administered agents (Figure 1) included aminoglycosides (46%), followed by colistin (26%), fosfomycin (21%), tigecycline (21%), and meropenem (13%).

CAZ-AVI was used empirically—without pathogen isolation or susceptibility testing—in 34% of patients (54/161). It was prescribed as first-line therapy in only 33% of cases (53/161). The median duration of treatment was 10 days (IQR 7–14). Patients receiving CAZ-AVI had been hospitalized for a median of 29 days (IQR 16–43) prior to infection onset.

Forty-two percent of patients (68/161) had a clinical failure.

The 30-day mortality rate was 24% (38/161). Relapse or persistent infection within 30 days after CAZ-AVI discontinuation occurred in 21% of cases, with resistance to CAZ-AVI emerging in 12% of isolates.

Clinical failure was significantly associated with a higher median SOFA score at admission (*p* = 0.02), liver transplantation (*p* = 0.008), and the development of acute conditions during ICU stay (*p* = 0.003), particularly septic shock (*p* = 0.003) requiring vasopressor support (*p* = 0.003) and acute renal failure (*p* < 0.0001). Additional risk factors included the need for renal replacement therapy (*p* = 0.001), ongoing steroid therapy (*p* = 0.002), immunosuppressive treatment (*p* = 0.029), and prolonged ICU stay (*p* = 0.034). Furthermore, the emergence of CAZ-AVI resistance was significantly associated with clinical failure (*p* < 0.0001). Conversely, patients admitted to the ICU due to polytrauma tended to have better outcomes (*p* = 0.032). The site of infection was not associated with clinical outcome (Table 2).

Table 3 describes the cases of CAZ-AVI resistance. Specifically, 15 strains were identified as *K. pneumoniae* and 3 as *P. aeruginosa*. Three strains exhibited CAZ-AVI resistance at the time of first isolation. Notably, 8 out of 11 relapsed or persistent strains regained susceptibility to meropenem following CAZ-AVI treatment.

Clinical breakpoints according to EUCAST 2023 were defined as follows: for CAZ-AVI, susceptible ≤ 8 mg/L and resistant > 8 mg/L (applicable to both *Enterobacterales* and *P. aeruginosa*); for meropenem, susceptible ≤ 2 mg/L and resistant > 8 mg/L (also for both *Enterobacterales* and *P. aeruginosa*).

In the multivariate logistic regression analysis, only steroid therapy [OR 5.601 (95% CI: 1.949–16.001), *p* = 0.001], the onset of acute renal failure [OR 3.632 (95% CI: 1.051–12.546), *p* = 0.041], and septic shock [OR 2.654 (95% CI: 1.086–6.484), *p* = 0.032] were independently associated with a negative outcome (Table 4).

Among the 14 solid organ transplant (SOT) recipients treated with CAZ-AVI in ICUs, 1 was a kidney transplant recipient and 13 were liver transplant recipients. As shown in Table 5, 77% of the liver transplant group were male, with a median age of 59 years. The median CCI was 3, and the median MELD-Na (Model for End-Stage Liver Disease-Sodium) score prior to transplantation was 23. Immunosuppressive therapy included corticosteroids and tacrolimus (69%), cyclosporine (23%), and tacrolimus combined with everolimus (8%). At ICU admission, the median APACHE II score was 12 and the median SOFA score was 7. Among these patients, 77% required mechanical ventilation, 38% required vasopressors, and 31% underwent renal replacement therapy as part of life-supporting treatment. None required ECMO. Most patients developed early (38%) or intermediate (46%) infections, while late infections occurred in only 6%. Ventilator-associated pneumonia was the most common infection (46%), followed by bloodstream infections (31%) and complicated urinary tract infections (15%). CAZ-AVI was used empirically in 38% of cases and as part of combination therapy in 23%. The median duration of therapy was 9 days.

As previously mentioned, most liver transplant recipients experienced a negative outcome (*p* = 0.034) in terms of both mortality and clinical failure. Therefore, Table 6 compares the characteristics of liver transplant recipients with those of other patients treated with CAZ-AVI. Liver transplant recipients were younger (*p* = 0.018), had undergone surgical intervention within the previous 6 months (*p* = 0.007), and were more frequently admitted from surgical departments (*p* < 0.001). They also underwent surgical drainage more often (*p* = 0.022), had longer hospital stays (*p* < 0.001), and had more frequently received other antimicrobial therapies within one month prior to infection onset (*p* = 0.02). Most were infected with *K. pneumoniae* (*p* = 0.01), particularly KPC-producing (*p* = 0.016) or multidrug-resistant (MDR) strains (*p* = 0.027), whereas patients in the other group were more frequently affected by *P. aeruginosa* infections (*p* = 0.01). Liver transplant recipients also developed acute renal impairment more frequently (*p* = 0.005) and experienced microbiological failure with relapse or persistent infection (*p* = 0.006). None of them had severe SARS-CoV-2 infection (*p* = 0.005).

## 3. Discussion

This observational cohort study retrospectively evaluated severe Gram-negative bacterial infections treated with ceftazidime–avibactam in critically ill patients admitted to four different Intensive Care Units in the Marche region (Italy) over a period of nearly four years. The primary aim was to assess the efficacy of this novel antimicrobial agent, and the secondary aim was to identify risk factors associated with poor clinical outcomes. CAZ-AVI was successfully used in 58% of cases, with an overall mortality rate of 24%. These findings are consistent with previously published studies, which reported success rates ranging from 54% to 71% and 30-day mortality rates between 25.3% and 32.4% [13,14,15,16,17,18]. However, clinical failure occurred in 42% of cases. Notably, relapse or persistent infection was observed in 25% (41/161) of patients. Similar rates of clinical failure have been reported in recent studies. For example, a multicenter observational study conducted between 2016 and 2017 involving 138 inpatients treated for KPC-producing *K. pneumoniae* infections reported an 8.7% relapse rate following CAZ-AVI discontinuation [14], while Shield et al. documented a 23% relapse rate within 90 days [19]. Krapp et al. reported a 33% failure rate, with relapse occurring in 2 out of 6 patients. This subgroup had a mean CCI of 4 and a mean SOFA score of 6 at the time of culture collection; five had severe renal dysfunction and four required hemodialysis. Most were septic, and two patients with carbapenem-resistant *K. pneumoniae* died [20]. These factors likely contributed to poor outcomes, as confirmed by larger cohort studies [13,21,22], not only due to increased mortality risk but also because they may alter the pharmacokinetic and pharmacodynamic properties of CAZ-AVI.

In this study, the development of CAZ-AVI resistance was observed in 26% of relapsed or persistent strains. This finding is clinically relevant and underscores the importance of the appropriate use of this antibiotic. Shortly after its introduction to the market, several studies reported outbreaks of CAZ-AVI-resistant strains. In 2015, the first case of CAZ-AVI resistance in a *K. pneumoniae* isolate from a patient with no prior exposure to CAZ-AVI was published, highlighting the need to test CAZ-AVI susceptibility in isolates that are KPC-positive and MBL-negative [23]. Previous studies have shown that resistance to CAZ-AVI often arises following prolonged treatment [24], and that pneumonia and renal replacement therapy are independent risk factors for microbiological failure and resistance emergence [25], likely due to suboptimal drug exposure at the site of infection. In conclusion, these findings support the notion that inappropriate dosing or extended treatment duration may contribute to the development of resistance—an explanation that may also apply to the present study.

Second, life-supporting therapies such as continuous renal replacement therapy (CRRT) and extracorporeal membrane oxygenation (ECMO) can significantly alter antimicrobial clearance. In the case of CRRT, factors such as the filter material, dialysate flow, and ultrafiltration rate must be considered to avoid underdosing or overdosing. While dosing nomograms exist for CRRT, far less is known about optimal dosing during ECMO [26]. Similarly, augmented renal clearance—frequently observed in critically ill patients with normal serum creatinine—poses a risk of underexposure. In this study, clinical failure and high resistance rates may reflect suboptimal antimicrobial dosing.

Moreover, pharmacokinetic/pharmacodynamic (PK/PD) studies suggest that prolonged or continuous infusion (CI) of CAZ-AVI may help achieve aggressive PK/PD targets in patients undergoing CRRT with residual renal function. This was supported by a study by Gatti et al., involving eight critically ill patients treated with continuous infusion of CAZ-AVI. Their findings suggest that initiating therapy with a full-dose CI regimen allows for the rapid attainment of optimal PK/PD targets within the first 72 h. Subsequent dose adjustment to 1.25 g every 8 h over 8 h, guided by real-time TDM, may help maintain these targets [27]. However, despite various proposed strategies, standardized administration protocols are still lacking.

Finally, other contributing factors may include heterogeneity in antimicrobial regimens, variability in pathogen profiles, and delays in initiating CAZ-AVI therapy.

In the study population, most critically ill patients with a negative outcome developed acute complications during their ICU stay, such as septic shock—confirmed by multivariate analysis as an independent risk factor for poor clinical outcome—requiring vasopressor support, and acute renal failure—also identified as an independent risk factor—requiring renal replacement therapy in the majority of cases. It is well established that septic shock is associated with poor outcomes, both in terms of mortality and microbiological failure. In patients with severe sepsis and septic shock, extensive fluid extravasation into the interstitial space due to endothelial damage and capillary leakage leads to increased interstitial volume. This condition may result in subtherapeutic antibiotic concentrations precisely when more intensive antimicrobial regimens are required [28]. Therefore, during septic shock, antimicrobial therapy should be optimized using dosing strategies based on pharmacokinetic/pharmacodynamic (PK/PD) principles and the specific properties of each drug [29]. It is not surprising that acute renal failure and the need for renal replacement therapy are independent risk factors for clinical failure, as these conditions not only worsen outcomes in critically ill patients but also increase the risk of inadequate antibiotic exposure.

The rate of treatment failure was higher among patients who were immunosuppressed due to both corticosteroid therapy and other immunosuppressive agents, although only corticosteroid use was independently associated with a negative outcome. It is well known that severe infections in immunocompromised patients are associated with poor prognosis, likely due to impaired pathogen clearance. A multicenter observational study involving 68 critically ill patients also found a correlation between prior corticosteroid use and adverse outcomes [13]. However, the use of corticosteroids in critically ill patients remains controversial. Some studies support their use in specific contexts, such as septic shock. For instance, a recent meta-analysis found that systemic corticosteroids may accelerate shock resolution [30], and the Surviving Sepsis Campaign recommends intravenous corticosteroids for adults with septic shock requiring ongoing vasopressor therapy, albeit as a weak recommendation with moderate-quality evidence [25]. Nonetheless, the optimal duration and dosing regimen remain uncertain. In the present study, prior or chronic corticosteroid use was considered a marker of immunosuppression.

Lastly, no statistically significant differences were observed between patients treated with combination therapy and those receiving monotherapy. This finding is supported by at least 2 meta-analyses, which included a total of 503 patients from 7 randomized controlled trials and 6 retrospective studies, and 11 studies, respectively [31,32]. Given the potential toxicity associated with combination therapy, monotherapy with ceftazidime–avibactam should be preferred—provided that the pathogen is susceptible to CAZ-AVI and adequate drug concentrations above the MIC can be achieved at the site of infection. This approach is also recommended in the most recent guidelines for the treatment of multidrug-resistant Gram-negative bacteria. The only clear indication for combination therapy remains infections caused by metallo-β-lactamase (MBL)-producing organisms, where the addition of aztreonam is essential to achieve clinical cure [33,34].

Given the poor outcomes observed among liver transplant recipients, a sub-analysis was conducted to further investigate this subgroup. Compared to the rest of the study population, liver transplant recipients had significantly worse outcomes (*p* = 0.008), including a higher rate of relapse or persistent infection (*p* = 0.006). However, data in the literature on this topic remain limited or are based on small-scale studies.

It is well established that SOT recipients are more vulnerable to infections due to underlying conditions, prolonged hospitalizations, pre- and post-transplant immunosuppression, frequent antibiotic exposure [35], and the use of invasive devices. These factors place them at particularly high risk for colonization and infection by MDR organisms. Among the SOT recipients, liver transplant patients are especially prone to Gram-negative bacterial infections due to the technical complexity of the surgical procedure, prolonged operative times, increased transfusion requirements, extended ICU stays, and overall longer hospitalizations [36].

The use of CAZ-AVI for CRE infections in SOT recipients has not been systematically evaluated. Data on its efficacy in liver transplant recipients are limited and somewhat conflicting. Some small-scale studies have shown promising results—for example, a retrospective study by Chen F. et al. [37] reported a 30-day mortality rate of 38.1%, comparable to that observed in the general population. However, a more recent study involving 81 inpatients identified SOT as an independent risk factor for poor outcomes, likely due to frequent surgical complications, prolonged hospitalizations, and polymicrobial infections [17].

In the present study, a concerning finding was the high rate of relapse or persistent infections and the emergence of CAZ-AVI resistance in second isolates from liver transplant recipients.

Although these findings suggest a poor outcome of CAZ-AVI treatment in SOT recipients, larger studies are needed to validate these observations. Notably, most infections in this group occurred during the early (38%) and intermediate (46%) post-transplant periods, when patients are more clinically and immunologically compromised. However, the quality of evidence remains low due to the lack of large-scale studies and clinical trials.

Consequently, the American Society of Transplantation Infectious Diseases (AST) recommends the use of CAZ-AVI for the treatment of CRE infections, when susceptibility is confirmed, and for MDR or XDR *P. aeruginosa* infections [38].

This study has several limitations and potential sources of bias that should be acknowledged. The retrospective design inherently introduces the risk of confounding factors that may have influenced the observed outcomes. Prospective studies are needed to validate the risk factors identified for clinical failure. The relatively small sample size limits the generalizability of the findings, particularly for subgroup analyses such as those involving liver transplant recipients. Expanding the cohort in future research would help strengthen the robustness of the conclusions. Additional investigations are also warranted to explore hypotheses such as the genetic characteristics of resistant strains. Moreover, the diagnosis of ventilator-associated pneumonia (VAP)—the most common infection in this cohort—is particularly challenging in critically ill patients, as clinical, radiological, and microbiological findings often overlap with other conditions such as pulmonary edema, atelectasis, or ARDS. This raises the risk of misclassification bias. Selection bias is also possible, as the inclusion criterion of receiving at least 72 h of CAZ-AVI may have excluded patients with early mortality or rapid clinical improvement. Confounding by indication may have occurred, given that CAZ-AVI was more likely used in patients with more severe or resistant infections. Additionally, the heterogeneity of infection types (e.g., VAP, bloodstream infections, and cUTIs) may limit the applicability of findings to specific clinical scenarios. The inclusion of a small subgroup of liver transplant recipients (<10% of the cohort) further introduces the risk of underpowered comparisons and overinterpretation. Finally, the empirical use of CAZ-AVI in a substantial proportion of patients without microbiological confirmation may have introduced information bias, complicating the attribution of outcomes to the drug’s efficacy.

## 4. Materials and Methods

A multicenter, cohort retrospective study was conducted in four Intensive Care Units located in three hospitals in Marche, Italy. The study setting consisted of the following: the Regional University Hospital in Ancona, “Azienda Ospedaliero-Universitaria delle Marche,” a tertiary referral center with two separate ICUs, and “Azienda Ospedaliera Marche Nord,” which includes the following two secondary-level hospitals: “Santa Croce” Hospital in Fano and “San Salvatore” Hospital in Pesaro. All adult critically ill patients admitted to these ICUs from January 2020 to September 2023 who received at least 72 h of CAZ-AVI for the treatment of a confirmed or suspected GNB infection were included in the study.

Data collection and general definitions: Medical data were collected from patient records. The report of patients treated with CAZ-AVI during the study period was provided by the corresponding hospital pharmacies. For each patient, the following data were recorded: sex, age; chronic comorbidities [diabetes mellitus, chronic obstructive pulmonary disease (COPD), cardiovascular, neurological, hematological, oncological, and gastrointestinal disorders, chronic kidney disease (CKD), and causes of immunosuppression such as HIV, solid organ transplantation, or bone marrow transplantation], and CCI [39]; reasons for ICU admission: postoperative care, acute respiratory failure, polytrauma, sepsis/septic shock, other (necrotic hemorrhagic pancreatitis, hepatic decompensation, stroke, status epilepticus, cardiocirculatory arrest, and encephalitis); previous surgery (within 6 months before the infection), corticosteroid and/or immunosuppressive therapy (within one month before the infection, including chronic immunosuppressive therapy); prior CRE intestinal colonization (confirmed by a rectal swab performed within 6 months prior to ICU admission), previous hospitalizations (longer than one month), and antimicrobial therapy exposure (within one month prior to infection); acute comorbidities developed during ICU stay [septic shock, acute kidney injury (AKI), acute respiratory distress syndrome (ARDS), pulmonary embolism, deep vein thrombosis, bleeding, stroke, acute myocardial infarction, and other conditions such as epilepsy, heart failure, intestinal perforation, atrial fibrillation, intestinal obstruction, pneumothorax, and cytomegalovirus (CMV) reactivation] were investigated

Moreover, life-supporting therapies, including mechanical ventilation, renal replacement therapy, extracorporeal membrane oxygenation (ECMO), the use of vasopressors, and the presence of devices within 72 h prior to the onset of infection, were assessed as pre-infection variables. Infection-related variables included the following: the site and severity of the infection (pneumonia, complicated urinary tract infection, complicated intra-abdominal infection, or other), and whether sepsis or septic shock was present. The causative pathogen and its resistance profile were recorded when the infection was microbiologically confirmed. For the purposes of this study, all susceptibility patterns were reviewed and classified according to the guidelines of the European Society of Clinical Microbiology and Infectious Diseases (ESCMID). Specifically, multidrug resistance (MDR) was defined as non-susceptibility to at least one agent in three or more antimicrobial categories; extensive drug-resistance (XDR) as non-susceptibility to at least one agent in all but two or fewer antimicrobial categories (i.e., bacterial isolates remain susceptible to only one or two categories); and pan-drug resistance (PDR) as non-susceptibility to all agents in all antimicrobial categories [40]. Regarding *P. aeruginosa*, since 2018, a new classification has been introduced: “difficult-to-treat resistance” (DTR), defined as non-susceptibility to all of the following agents: ceftazidime, cefepime, piperacillin–tazobactam, imipenem–cilastatin, meropenem, ciprofloxacin, levofloxacin, and aztreonam [41]. The definitions of MDR and XDR *P. aeruginosa* are consistent with those described above.

Concerning the treatment, the following variables were assessed: duration of therapy, whether CAZ-AVI was used as part of a combination therapy regimen or administered empirically (i.e., without susceptibility testing available); its use as first-line therapy or as salvage therapy, following at least one prior therapeutic line. Clinical outcome was considered positive in cases of clinical improvement and survival 30 days after treatment, and negative in cases of death and/or relapse or persistent infection within 30 days. The dosing regimen of ceftazidime/avibactam was determined based on clinical judgment.

The severity of underlying diseases was evaluated upon ICU admission using the Sequential Organ Failure Assessment (SOFA) score and the Acute Physiology and Chronic Health Evaluation II (APACHE II) score [42,43]. If the patient was already sedated, the Glasgow Coma Scale (GCS) was estimated as the score the patient would have in the absence of sedative medications.

Regarding liver transplant recipients, the pre-transplantation MELD-Na score was calculated. Infections occurring within the first month after transplantation were defined as “early,” those between one and six months as “intermediate,” and those after six months as “late” [44].

The primary objective was to evaluate the clinical outcome 30 days after the onset of infection.

Clinical failure was defined when the following criteria were present: (1) the patient died within 30 days of the onset of infection; (2) persistence of signs and symptoms of infection and/or persistently positive cultures; (3) recurrence of the infection within 30 days after CAZ-AVI discontinuation. Patients who did not meet any of these criteria were considered to have a successful clinical outcome.

Microbiological Data: The strains isolated from clinical specimens of patients admitted to the ICUs of the hospital in Ancona were identified using matrix-assisted laser desorption/ionization time-of-flight mass spectrometry (MALDI-TOF/MS). The presence of KPC was detected using the GeneXpert system (Cepheid, Sunnyvale, CA, USA). Susceptibility testing was performed with the Vitek 2 System and interpreted according to the 2023 EUCAST guidelines [45], with the exception of ceftazidime–avibactam susceptibility, which was determined using the MIC Test Strip (Liofilchem, Roseto degli Abruzzi, Italy).

Statistical Analysis: Quantitative data are expressed as medians with interquartile ranges (IQR: 1st and 3rd quartiles) and were compared using the Mann–Whitney U test (for non-normally distributed variables) or Student’s *t*-test (for normally distributed variables). Qualitative variables are presented as absolute frequencies and relative percentages, and were compared using the Chi-square test or Fisher’s exact test, as appropriate. Factors associated with clinical failure were analyzed using a stepwise binary logistic regression model, in which variables found to be significant at the univariate level (*p* < 0.05) were included and further examined through multivariate logistic regression analysis to identify independent risk factors for negative outcomes. Statistical analyses were performed using SPSS for Windows, version 20 (SPSS Inc., Chicago, IL, USA).

## 5. Conclusions

Although this study has several limitations, it allowed for some important considerations. The treatment of MDR GNB presents many challenges, especially in ICU settings. Ceftazidime–avibactam is the first member of a new generation of β-lactam/β-lactamase inhibitor combinations, with proven safety and efficacy profiles for infections caused by multidrug-resistant Gram-negative pathogens, particularly CRE and *P. aeruginosa*.

With a crude mortality rate of 24%, this study confirmed that the mortality rate among patients treated with this antimicrobial in ICUs is comparable to that reported in other studies focusing on heterogeneous populations. However, the rapid emergence of resistance to this agent, despite its recent introduction into the antimicrobial armamentarium, highlights the need for ongoing surveillance and the implementation of antimicrobial stewardship strategies to preserve one of the last therapeutic options against MDR GNB.

Further studies involving larger cohorts of critically ill patients are needed to confirm these findings, particularly regarding the optimal dosing regimen for this patient population, with special attention to the most vulnerable groups, such as SOT recipients.

## Figures and Tables

**Figure 1 antibiotics-14-00797-f001:**
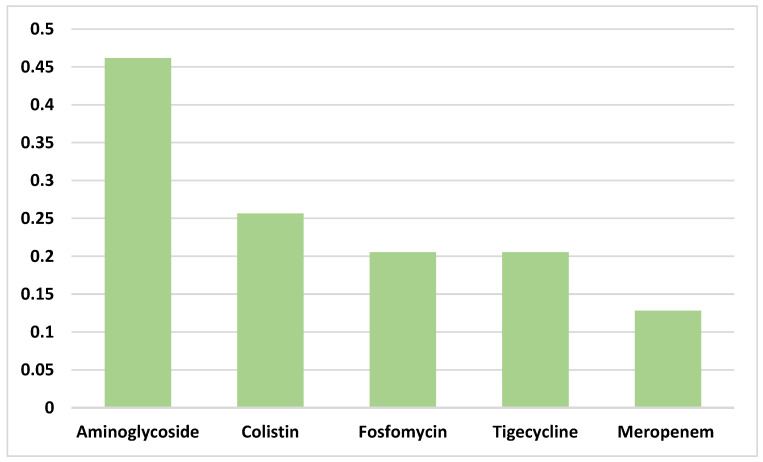
Combination therapy with CAZ-AVI.

**Table 1 antibiotics-14-00797-t001:** Microbiological characteristics of main pathogens treated with CAZ-AVI.

%	n	Variable
		Microbiological data
44%	71	*Klebsiella pneumoniae*
34%	54	*Pseudomonas aeruginosa*
1%	1	*Enterobacter cloacae*
4%	6	*Escherichia coli*
1%	2	*Proteus mirabilis*
3%	5	*Klebsiella aerogenes*
1%	2	*Klebsiella oxytoca*
1%	1	*Acinetobacter baumannii*
1%	1	*Morganella morganii*
1%	1	*Serratia marcescens*
27%	44	Polymicrobial infection
58%	94	Monomicrobial infection
		Resistance pattern
		*Enterobacterales*
6%	9	non-MDR
44%	71	MDR
37%	59	KPC carbapenemase
1%	1	OXA-48 carbapenemase
8%	13	ESBL
2%	5	XDR
		*P. aeruginosa*
1%	2	DTR
7%	11	MDR
1%	2	XDR
24%	39	Non-MDR
		*A. baumannii*
1%	1	XDR

**Table 2 antibiotics-14-00797-t002:** Univariate analysis of factors associated with clinical failure of CAZ-AVI treatment.

	Clinical Failure (n = 68)	Successful Clinical Outcome (n = 93)	All (161)	
*p*-Value	n (%)	n (%)	n (%)	Variable
				Patient-related variables
0.617	45 (66%)	65 (70%)	110 (68%)	Male
	23 (34%)	28 (30%)	51 (32%)	Female
0.968	65.5 (54.5–73)	64 (55.5–74)	64 (55.5–73)	Age (years), mean (IQR)
0.407	46 (68%)	57 (61%)	103 (64%)	Charlson’s comorbidity index > 3
0.721	17 (25%)	21 (23%)	38 (24%)	Apache II score > 15
0.02	6 (4.25–7)	5 (3–7)	5 (4–7)	SOFA score, median (IQR)
				Comorbidities
0.143	11 (16%)	24 (26%)	35 (22%)	Diabetes mellitus
0.699	11 (16%)	13 (14%)	24 (15%)	COPD
0.516	4 (6%)	8 (9%)	12 (8%)	Hematological disorders
0.683	4 (6%)	7 (8%)	11 (7%)	Solid tumors
0.576	35 (52%)	55 (56%)	87 (54%)	Cardiovascular disorders
0.863	11 (16%)	16 (17%)	27 (17%)	Neurological diseases
0.799	12 (18%)	15 (16%)	27 (17%)	GI disorders
0.008	10 (15%)	3 (3%)	13 (8%)	Liver transplant recipients
0.004	59 (87%)	62 (67%)	121 (75%)	Acute diseases
0.003	33 (49%)	24 (26%)	57 (35%)	Septic shock
<0.0001	23 (34%)	9 (10%)	32 (20%)	Acute renal failure
0.225	50 (74%)	60 (65%)	110 (68%)	Admission in ICU from other departments
				Pre-infection variables
0.211	58 (85%)	72 (77%)	130 (81%)	Central venous catheter
0.800	21 (18%)	27 (29%)	48 (30%)	Surgical drainage ^a^
0.569	18 (27%)	21 (23%)	39 (24%)	Tracheostomy
0.892	59 (87%)	80 (86%)	139 (86%)	Bladder catheter
0.942	56 (82%)	77 (83%)	133 (83%)	Mechanical ventilation ^a^
0.001	19 (28%)	8 (9%)	27 (17%)	Renal replacement therapy
0.003	33 (49%)	24 (26%)	57 (35%)	Vasopressors
0.001	30 (44%)	18 (19%)	48 (30%)	Corticosteroid therapy ^b^
0.029	15 (22%)	9 (10%)	24 (15%)	Immunosuppressive therapy ^b,c^
0.286	25 (37%)	42 (45%)	67 (42%)	Previous surgery ^d^
0.592	38 (56%)	48 (52%)	86 (53%)	Hospitalization more than a month before the infection
0.584	15 (22%)	24 (26%)	39 (24%)	Positive rectal swab for CRE at the admission or within 6 months before
				Reason for ICU admission
0.241	7 (10%)	5 (5%)	12 (8%)	Sepsis
0.720	23 (33%)	34 (37%)	57 (35%)	Postoperative care
0.896	22 (32%)	31 (33%)	53 (33%)	Acute respiratory failure
0.032	5 (7%)	18 (19%)	23 (14%)	Polytrauma
0.086	12 (17%)	8 (9%)	20 (12%)	Others **
0.034	34 (21–43.75)	26 (14–42.5)	29 (16–43)	Days in ICU before the onset of infection
				Infection variables
0.383	4 (6%)	9 (10%)	13 (8%)	Polymicrobial
				Site of isolation
0.074	7 (11%)	17 (23%)	24 (17%)	Urinary tract
0.144	52 (82%)	54 (72%)	106 (77%)	Bronchial/pleural fluid
0.688	2 (3%)	4 (5%)	6 (4%)	Abdominal fluid
0.827	3 (5%)	3 (4%)	6 (4%)	Wounds
0.683	12 (19%)	12 (16%)	24 (17%)	Blood
				Microbiological variables
0.354	35 (52%)	41 (44%)	76 (47%)	*Klebsiella pneumoniae*
0.723	23 (34%)	29 (31%)	52 (32%)	*Pseudomonas aeruginosa*
0.806	4 (6%)	6 (7%)	10 (6%)	Others ***
0.092	30 (44%)	29 (31%)	59 (36%)	KPC
0.567	5 (8%)	8 (11%)	13 (10%)	ESBL
0.215	41 (65%)	41 (55%)	82 (60%)	MDR
0.134	5 (7%)	2 (2%)	7 (4%)	XDR

Data are expressed as No. (%) unless otherwise specified. Abbreviations: IQR, interquartile range; COPD, chronic obstructive pulmonary disease. ^a^ During the 72 h preceding BSI onset; ^b^ During the 30 days preceding BSI onset; ^c^ Excluding therapy with steroids; ^d^ During the 3 months preceding BSI onset. ** Others: necrotic hemorrhagic pancreatitis, hepatic decompensation, stroke, status epilepticus, cardiocirculatory arrest, encephalitis. *** Others: *P. mirabilis*, *M. morganii*, *S. marcescens*, *K. aerogenes*, *K. oxytoca*, *E. cloacae*, *E. coli*, *A. baumannii*.

**Table 3 antibiotics-14-00797-t003:** MIC values for meropenem and CAZ-AVI, both for the first isolates and for the relapse/persistent strains.

Relapse/Persistent Infection		First Infection		Pathogen	Year of Isolation	Case
MIC CAZ-AVI (mg/L)	MIC Meropenem (mg/L)	MIC CAZ-AVI (mg/L)	MIC Meropenem (mg/L)			
>16	8	6	>16	*K. pneumoniae*	2020	#1
>16	1.5	>16	>16	*K. pneumoniae*	2020	#2
>16	>16	8	>16	*K. pneumoniae*	2020	#3
>16	>16	8	>16	*K. pneumoniae*	2020	#4
>16	1	4	>16	*K. pneumoniae*	2020	#5
>16	>16	>16	>16	*K. pneumoniae*	2021	#6
>16	>16	>16	>16	*K. pneumoniae*	2021	#7
>16	>16	1	>16	*K. pneumoniae*	2021	#8
>16	1	2	>16	*K. pneumoniae*	2021	#9
>16	>16	4	>16	*K. pneumoniae*	2021	#10
>16	≤0.25	8	>16	*K. pneumoniae*	2021	#11
>16	>16	2	>16	*P. aeruginosa*	2021	#12
>16	≤0.25	8	>16	*K. pneumoniae*	2021	#13
>16	8	2	4	*P. aeruginosa*	2022	#14
>16	1	4	>16	*K. pneumoniae*	2023	#15
>16	1	4	>16	*K. pneumoniae*	2023	#16
>16	0.5	2	1	*P. aeruginosa*	2023	#17
>16	>16	4	>16	*K. pneumoniae*	2023	#18

**Table 4 antibiotics-14-00797-t004:** Risk factors independently related to 30-day negative outcome.

*p*-Value	OR (95% CI)	Variable
0.001	5.601 (1.949–16.001)	Corticosteroid therapy
0.041	3.632 (1.051–12.546)	Acute renal failure
0.032	2.654 (1.086–6.484)	Septic shock

**Table 5 antibiotics-14-00797-t005:** Clinical features of critically ill liver transplant recipients.

%	n	Variables
77%	10	Male sex
53–62	59	Age (years) median, IQR
13–31	23	MELD-Na pre-OLT: median (IQR)
2–4	3	Charlson’s comorbidity index: median (IQR)
		Immunosuppressive therapy
69%	9	Tacrolimus
8%	1	Tacrolimus + everolimus
23%	3	Cyclosporin
38%	5	Early post-transplantation infection
46%	6	Intermediate post-transplantation infection
15%	2	Late post-transplantation infection
		Type of infection
31%	4	BSI
8%	1	CLABSI
8%	1	HAP
46%	6	VAP
15%	2	cUTI
8%	1	cIAI
8%	1	Suspected MDR GNB infection
8%	1	Other (surgical site infection)
		Life-supporting therapies
77%	10	Mechanical ventilation
31%	4	CVVH
38%	5	Vasopressors
		Severity scores: median (IQR)
12–14	12	Apache II score at admission
7–10	7	SOFA score at admission
7–14	9	Duration of therapy, median (IQR)
38%	5	Empiric use of CAZ-AVI
23%	3	Combination therapy
31%	4	CAZ-AVI as first line therapy

Data are expressed as No. (%) unless otherwise specified. Abbreviations: IQR, interquartile range; BSI, bloodstream infection; CLABSI, central line-associated bloodstream infection; HAP, hospital-acquired pneumonia; VAP, ventilator-associated pneumonia; cUTI, complicated urinary tract infection; cIAI, complicated intra-abdominal infection; MDR GNB, multi-drug resistant Gram-negative bacteria; CVVH, continuous veno-venous hemofiltration; CAZ-AVI, ceftazidime–avibactam.

**Table 6 antibiotics-14-00797-t006:** Demographic, clinical, and microbiological features of liver transplant recipients compared with other critically ill patients.

	Other Patients (n = 148)	Liver Transplant Recipients (n = 13)	
*p*-Value	n (%)	n (%)	Variable
0.018	66.5 (56–74)	59 (52.5–62)	Age, median (IQR)
0.005	54 (37%)	0	COVID-19 disease
0.007	57 (39%)	10 (77%)	Previous surgery
<0.001	73 (49%)	13 (100%)	Hospitalization more than a month before the infection
0.020	104 (70%)	13 (100%)	Previous antibiotic therapy (within one month of infection)
<0.001	17 (12%)	9 (69%)	Admission in ICU from surgical departments
<0.001	44 (30%)	13 (100%)	ICU admission for postoperative care
0.005	53 (36%)	0	ICU admission for acute respiratory failure
0.001	64 (43%)	12 (92%)	*Klebsiella pneumoniae* infection
0.01	52 (35%)	0	*Pseudomonas aeruginosa* infection
0.016	50 (34%)	9 (69%)	KPC
0.027	71 (56%)	11 (92%)	MDR
0.022	40 (27%)	8 (62%)	Surgical drainage ^a^
<0.0001	35 (24%)	13 (100%)	Steroid therapy ^b^
<0.0001	11 (7%)	13 (100%)	Immunosuppressive therapy ^b,c^
0.005	25 (17%)	7 (54%)	Acute renal failure
0.006	26 (18%)	7 (54%)	Relapse
0.008	58 (39%)	10 (77%)	Outcome negative
0.07	28 (16–42)	43 (24–85)	Days in ICU before the onset of infection, median (IQR)
0.0002	5 (3–7)	7 (7–10.5)	SOFA score, median (IQR)

Data are expressed as No. (%) unless otherwise specified. Abbreviations: IQR, interquartile range; ICU, intensive care unit; MDR, multi drug resistance. ^a^ During the 72 h preceding BSI onset; ^b^ During the 30 days preceding BSI onset; ^c^ Excluding therapy with steroids.

## Data Availability

Data were collected from the medical case sheets and the laboratory and radiology data available on the hospital’s electronic database.

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
