# Peer review of "Ceftazidime–Avibactam in Critically Ill Patients: A Multicenter Observational Study"

_antibiotics, 2025, doi:10.3390/antibiotics14080797_

Round 1

Reviewer 1 Report

Comments and Suggestions for Authors

The study reporting the usage and outcomes of therapy with ceftazidime-avibactam in critically ill patients across 4 ICUs in 3 hospitals in Italy addresses an important topic of interest. There is not much data on clinical use of this relatively new drug mainly approved for MDR game negative infections. The study has been conducted very well and the authors have presented all the pertinent data in a comprehensive manner. Few queries are:

Abstract: Results-it is mentioned “Among a total of 161 patients, this novel antimicrobial molecule was successfully used in 58% (n=93/161)”; what do the authors mean by saying “successfully used”?, plz write in scientific terms.

Results:

Line no. 178; How was the “successful clinical outcome” defined?

Line no. 179; How was “clinical failure” defined?

Line no. 184: what was meant by “therapeutic failure”; please define.

Line no. 217, 218: plz write full forms of CCI and MELD-Na at first instance.

The paper is too lengthy, particularly the results and discussion part. Few tables may be provided as supplementary files, if desired.  

Comments on the Quality of English Language

Minor to moderate editing in English language required.

Author Response

1 The study reporting the usage and outcomes of therapy with ceftazidime-avibactam in critically ill patients across 4 ICUs in 3 hospitals in Italy addresses an important topic of interest. There is not much data on clinical use of this relatively new drug mainly approved for MDR game negative infections. The study has been conducted very well and the authors have presented all the pertinent data in a comprehensive manner. Few queries are:

Abstract: Results-it is mentioned “Among a total of 161 patients, this novel antimicrobial molecule was successfully used in 58% (n=93/161)”; what do the authors mean by saying “successfully used”?, plz write in scientific terms. R: Done.

Results:

Line no. 178; How was the “successful clinical outcome” defined? R: Done.

Line no. 179; How was “clinical failure” defined? R: Done.

Line no. 184: what was meant by “therapeutic failure”; please define. R: Done.

Line no. 217, 218: plz write full forms of CCI and MELD-Na at first instance. R: done.

The paper is too lengthy, particularly the results and discussion part. Few tables may be provided as supplementary files, if desired.

Minor to moderate editing in English language required. R: We revised the article and improved the quality of the English.

Reviewer 2 Report

Comments and Suggestions for Authors

This study presents information of critically ill patients from 4 ICUs in Italy. -Information about this antimicrobial is needed and the study can provide more information on the use.

It is not clear if this is a cohort study, since it is not clearly stated. If so, it could be imporved by following and using the STROBE checklist.

Other possibilities to improve:

Observations:

  1. Title in capital letters, please correct.
  2. The abstract is incomplete, it does not provide adequate information about the study. Is it a cohort? How was the anlysis done? It does not provide any estimator of the relationship between the variables.
  3. Please refere to Enterobacterales instead of Enterobacteriaceae according to current nomenclature.
  4. Use italics for the microorganisms´ names. This recommendations is extensive to all the document.
  5. Please synthetize the introduction. Instead of 8 paragraphs about diverse items, please consider using just 3 pragraphs of the main topics of interest and focus on the resistance and potential use of ceftazidime.
  6. The aim is clear, however an objective in the liver transplant population was added. It is not clear the reason (since no information about was put in the introduction) and it seems that it was not a primary objective but one of chance. This group of patients are less than 10% of all the patients.
  7. Results: Please provide a table of the main characteristics of the patients, not a figure (figure 1). Please provide a figure of flow of the patients.
  8. Please provide a discriminated outcome: How many of the patients died, how many had a relapse and how many had a persistent infection.
  9. Figure 1 and figure 2 should be part of table one compiling the main characteristics of the included patients. Text can have the remaining data (not shown in the table) or to highlight important points of the data.
  10. Table 2,3, and 4 could be put in just one table, selecting the most important variables.
  11. Figure 3 and table 1 should be combined in one table with all the microbiological data (microorganisms and resistance patterns)
  12. The reason for the non-susceptibility to CAZAVI is not clear, since only a minority were P aeruginosa isolates and most isolate restores susceptibility to meropenem, and the number of isolates mith MBL was very low? Is there more microbiologic information (other MBL? other frequent mechanisms in the hospital?)
  13. Please, explore further on the possible biases that might be encountered. A lot of patients had ventilator associated pneumonia, which could be easily confused with a number of other conditions.
  14. Methods: It is not clear if this a cross sectional study or if it is a cohort. Please provide more information to be clear.
  15. Please provide working defintions for the pathologies included.
  16. Which methodology was chosen for carbapenemase resistance detection (agars?)?
  17. Please provide the working definition of relapse and persistent infection.
  18. The variables selected for the multivariate model were chosen according to the differences and p value. It is common to select a p value of p=0.2, instead of 0.05 for this step in the configuration of the model, could you explain why did you chose a different value?

Author Response

his study presents information of critically ill patients from 4 ICUs in Italy. -Information about this antimicrobial is needed and the study can provide more information on the use.

It is not clear if this is a cohort study, since it is not clearly stated. If so, it could be improved by following and using the STROBE checklist. R: Revised. We decided not to include a flowchart, as a very linear patient selection was adopted.

Other possibilities to improve:

Observations:

  1. Title in capital letters, please correct. R: Done.
  2. The abstract is incomplete, it does not provide adequate information about the study. Is it a cohort? How was the anlysis done? It does not provide any estimator of the relationship between the variables. R: we have clarified the aspects requested on the abstract.
  3. Please refere to Enterobacterales instead of Enterobacteriaceae according to current nomenclature. R: Done.
  4. Use italics for the microorganisms´ names. This recommendations is extensive to all the document. R: Done.
  5. Please synthetize the introduction. Instead of 8 paragraphs about diverse items, please consider using just 3 paragraphs of the main topics of interest and focus on the resistance and potential use of ceftazidime. R: Done.
  6. The aim is clear, however an objective in the liver transplant population was added. It is not clear the reason (since no information about was put in the introduction) and it seems that it was not a primary objective but one of chance. This group of patients are less than 10% of all the patients. R: Done.
  7. Results: Please provide a table of the main characteristics of the patients, not a figure (figure 1). Please provide a figure of flow of the patients. R: Since all data are available in the text and/or in Table 2, we removed Figure 1. We decided not to include a flowchart, as a very linear patient selection was adopted.
  8. Please provide a discriminated outcome: How many of the patients died, how many had a relapse and how many had a persistent infection. R: mortality rate and number of patients who had relapse or persistent infection is described, unfortunately the discrimination between relapse and persistence is not available so there isn’t a discriminated outcome.
  9. Figure 1 and figure 2 should be part of table one compiling the main characteristics of the included patients. Text can have the remaining data (not shown in the table) or to highlight important points of the data. R: Since all data are available in the text and/or in Table 2 and 4, we removed Figure 1 and 2.
  10. Table 2,3, and 4 could be put in just one table, selecting the most important variables. R: done.
  11. Figure 3 and table 1 should be combined in one table with all the microbiological data (microorganisms and resistance patterns) R: Done.
  12. The reason for the non-susceptibility to CAZ-AVI is not clear, since only a minority were P aeruginosa isolates and most isolate restores susceptibility to meropenem, and the number of isolates mith MBL was very low? Is there more microbiologic information (other MBL? other frequent mechanisms in the hospital?) R: unfortunately we don’t have the molecular resistance mechanism for all the isolates. It will be interesting to delve deeper with further studies on strains.
  13. Please, explore further on the possible biases that might be encountered. A lot of patients had ventilator associated pneumonia, which could be easily confused with a number of other conditions. R: Done.
  14. Methods: It is not clear if this a cross sectional study or if it is a cohort. Please provide more information to be clear. R: Done.
  15. Please provide working defintions for the pathologies included. R: we added the definitions below the tables
  16. Which methodology was chosen for carbapenemase resistance detection (agars?): R: Carbapenemase  genes were molecularly detected by Genexpert assay and resistance was confirmed by MICs determination through Vitek2 system.
  17. Please provide the working definition of relapse and persistent infection. R:Done.
  18. The variables selected for the multivariate model were chosen according to the differences and p value. It is common to select a p value of p=0.2, instead of 0.05 for this step in the configuration of the model, could you explain why did you chose a different value? R: we chose to use a p<0.05 because having an adequate sample size we wanted to achieve the greatest specificity of the result.

Reviewer 3 Report

Comments and Suggestions for Authors

Thank you for the opportunity to review the manuscript, "Ceftazidime/avibactam in critically ill patients: a multicentre observational study".

The manuscript adds to the evidence base for treatment of confirmed or suspected Gram-negative bacterial infections in a mixed patient group, using pragmatic methodology.

Key points:

  • Notable that in 26% of patients resistance was noted 
  • The study is strengthened by including 4 sites
  • Please clarify the primary result in the abstract, "successfully used". This is further explained in the main results section as:  Clinical outcome was considered positive in case of clinical improvement and survival after 30-days from the treatment
  • Liver transplant patients experienced worse outcomes
  • The results section spans four figures and nine tables. In some cases the results could be better explained in just text
Comments on the Quality of English Language
  • minor improvements are required to spelling and grammar throughout the manuscript

"succesfull", "fourty", "carcarbapenem", 

Author Response

Thank you for the opportunity to review the manuscript, "Ceftazidime/avibactam in critically ill patients: a multicentre observational study".

The manuscript adds to the evidence base for treatment of confirmed or suspected Gram-negative bacterial infections in a mixed patient group, using pragmatic methodology.

Key points:

  • Notable that in 26% of patients resistance was noted 
  • The study is strengthened by including 4 sites
  • Please clarify the primary result in the abstract, "successfully used". This is further explained in the main results section as: Clinical outcome was considered positive in case of clinical improvement and survival after 30-days from the treatment R: Done.
  • Liver transplant patients experienced worse outcomes
  • The results section spans four figures and nine tables. In some cases the results could be better explained in just text
    • minor improvements are required to spelling and grammar throughout the manuscript

    "succesfull", "fourty", "carcarbapenem", R: Done